# Isomadecassoside, a New Ursane-Type Triterpene Glycoside from *Centella asiatica* Leaves, Reduces Nitrite Levels in LPS-Stimulated Macrophages

**DOI:** 10.3390/biom11040494

**Published:** 2021-03-25

**Authors:** Giuseppina Chianese, Francesca Masi, Donatella Cicia, Daniele Ciceri, Sabrina Arpini, Mario Falzoni, Ester Pagano, Orazio Taglialatela-Scafati

**Affiliations:** 1Department of Pharmacy, School of Medicine and Surgery, University of Naples Federico II, Via D. Montesano 31, I-80131 Naples, Italy; g.chianese@unina.it (G.C.); francesca.masi@unina.it (F.M.); donatella.cicia@unina.it (D.C.); ester.pagano@unina.it (E.P.); 2INDENA SPA, Via Don Minzoni 6, 20090 Settala, Italy; daniele.ciceri@indena.com (D.C.); sabrina.arpini@indena.com (S.A.); mario.falzoni@indena.com (M.F.); 3Department of Pharmacy, University of Chieti G. D’Annunzio, Via dei Vestini, 66100 Chieti, Italy

**Keywords:** *Centella asiatica*, triterpenoid saponins, phytochemicals, anti-inflammatory activity

## Abstract

A madecassoside-rich fraction obtained from the industrial purification of *Centella asiatica* leaves afforded a new triterpene glycoside, named isomadecassoside (**4**), characterized by an ursane-type skeleton and migration of the double bond at Δ^20(21)^ in ring E. The structure of isomadecassoside was established by means of HR-ESIMS and detailed analysis of 1D and 2D NMR spectra, which allowed a complete NMR assignment. Studies on isolated J774A.1 macrophages stimulated by LPS revealed that isomadecassoside (**4**) inhibited nitrite production at non-cytotoxic concentrations, thus indicating an anti-inflammatory effect similar to that of madecassoside.

## 1. Introduction

*Centella asiatica* (L.) Urban, also known as gotu kola or Indian pennyworth, is a perennial herbaceous plant belonging to the family Apiaceae (Umbelliferae). It is native to the moist areas of tropical or subtropical regions of Southeast Asia such as India, Sri Lanka, China, Indonesia and Malaysia, but it is also diffused in South Africa and Madagascar [1]. *C. asiatica* has been employed since prehistoric times, and its leaves continue to be locally used nowadays as an antimicrobial agent, diuretic and as an anti-inflammatory agent, with special indication for gastric ulcers and syphilitic lesions. Topical preparations of *C. asiatica* are typically indicated to accelerate healing of skin ulcers and wounds, being able to promote the synthesis of collagen, normalize the hyperproliferation of keratinocytes and restore the natural homeostasis of the epidermis [1,2]. Phytochemical studies on this important medicinal plant have reported several biomolecules such as triterpenes (in the free and glycosylated forms) [3] and flavonoids [4], in addition to volatile components of the essential oils [5,6].

The most important bioactive constituents of *C. asiatica* are known as centelloids, pentacyclic triterpenoid (C_30_) saponins predominantly present in leaves and, to a lesser extent, in roots [7]. The structure of the sapogenin moiety can be traced back to two pentacyclic triterpenoid subtypes, the ursane and the oleanane series, that differ by the methyl substitution pattern at C-19 and C-20. The structural diversity within these series is given by the number and position of double bonds and the degree of hydroxylation and glycosylation, although the predominant members of the class are characterized by a Glucose (Glu)-Glucose (Glu)-Rhamnose (Rha) triglycoside esterifying the carboxylic group at C-28 [8]. The nomenclature of the several dozens of *Centella* saponins reported in the literature is sometimes confusing and lacks general rules. The most representative centelloids are asiaticoside (**1**) [9] and its C-6 hydroxylated analogue madecassoside (**2**) [10,11] (Figure 1), along with their corresponding aglycones asiatic acid and madecassic acid [12].

Due to its medicinal properties, exploited in several phytotherapeutic formulations on the market, interest in *C. asiatica* has increased over the years, and there have been significant results on the identification and chemical characterization of centelloids, as well as the determination of their biological profile in the pure form. In particular, asiaticoside (**1**) (at 20–80 µM) has been studied for its wound healing activity in normal as well as in diabetic rats, unveiling stimulating effects on collagen synthesis and promotion of fibroblast proliferation after a 10-day treatment [13]. The aglycone asiatic acid was found to exert a promising effect on SK-MEL-2 human melanoma cells with IC_50_ = 40 μM, due to the antiproliferative activity and induction of cell cycle arrest. In addition, asiatic acid has been also correlated to the cardiovascular protective effect of *C. asiatica* [14]. Madecassoside (**2**) and madecassic acid have shown anti-inflammatory activity since they can inhibit the production of lipopolysaccharide (LPS)-induced nitric oxide (NO) and prostaglandin E_2_ (PGE_2_) [15].

In the frame of our research project aimed at the detailed phytochemical characterization of medicinal plants used as dietary supplements, herein we describe the results of the analysis of a madecassoside-rich fraction obtained from *C. asiatica* leaves. This analysis led to the isolation of the new triterpenoid glycoside isomadecassoside (**4**) (Figure 2), along with the known madecassoside (**2**) [10,11] and terminoloside (also known as asiaticoside B, **3**) [11]. The anti-inflammatory activity of the isolated saponins **2**–**4**, measured by suppression of nitrite production in LPS-induced J774A.1 macrophage cells, is also reported.

## 2. Materials and Methods

### 2.1. General Experimental Procedures

Optical rotations (CH_3_OH) were measured at 589 nm on a P2000 Jasco (Dunmow, UK) polarimeter. Low and HR-ESIMS experiments were performed on a LTQ-Orbitrap mass spectrometer equipped with an ESI interface and Excalibur data system. ^1^H (700 MHz) and ^13^C (175 MHz) NMR spectra were measured on a Bruker Avance 700 spectrometer (Bruker^®^, Billerica, MA, USA). Chemical shifts are referenced to the residual solvent signal (CD_3_OD: δ_H_ 3.31, δ_C_ 49.0). Homonuclear ^1^H connectivities were determined by COSY (COrrelation SpectroscopY) experiments. Through-space ^1^H connectivities were evidenced using a NOESY (Nuclear Overhauser Enhancement SpectroscopY) experiment with a mixing time of 300 ms. One-bond heteronuclear ^1^H−^13^C connectivities was determined by the HSQC (Heteronuclear Single Quantum Correlation) experiment: two- and three-bond ^1^H−^13^C connectivities by gradient-HMBC (Heteronuclear Multiple Bond Correlation) experiments optimized for a ^2,3^*J* of 8 Hz. RP-HPLC-UV-vis separations were performed on a Shimadzu instrument, pump LC-10AD, equipped with an SPD-10A Detector, using Synergi 4u Polar-RP 80A (Phenomenex, Torrance, CA, USA) 250 × 4.60 mm column and a Rheodyne^®^ injector. Thin-layer chromatography (TLC) was performed on plates coated with silica gel 60 F254 Merck, 0.25 mm. Chemicals and solvents were from Merck Life Science S.r.l. (Milan, Italy) and were used without any further purification unless stated otherwise.

### 2.2. Extraction and Isolation

*Centella asiatica* leaves (300 kg, Madagascar origin, purchased by Indena) were poured in a percolator and extracted with 90% ethanol, at 70 °C, carrying out 5 extractions (about 27 V). The leachates were combined (dry residue about 90 kg), concentrated and adjusted with water and acetone to obtain a 50% dry residue and a 50% acetone solution, which was extracted with hexane (5 × 50 L) to separate the pentacyclic triterpene acids. The hydroacetonic layer, containing mainly the glycosides, was concentrated to obtain a soft mass of about 140 kg (dry residue about 70 kg), which was extracted with butanol (about 260 L). The butanol layers were combined (dry residue about 45 kg), concentrated and adjusted with water and methanol to have a 30% dry residue and a 60% methanolic suspension, which was solubilized by refluxing. The solution was cooled to 25 °C, and asiaticoside was crystallized under stirring for 24 h. The suspension was filtered to separate asiaticoside, and the mother liquors (madecassoside and terminoloside rich fraction, dry residue about 15 kg) were concentrated and adjusted with water and acetone to have a 40% dry residue, 22% acetone solution. This solution was loaded onto a reverse phase C18 Zeoprep column (500 kg). The elution was performed with water/acetone 78:22, collecting fractions of 200 kg each (flow rate 100 kg/h, column head pressure: 2.5–3 bar). The obtained fractions were combined as follows: head fractions up to the appearance of madecassoside, heart fractions (sum of madecassoside and terminoloside) and tails fraction. The head fractions were combined and concentrated to dryness, yielding 2.5 kg of an intermediate solid product. A part of this product (400 g) was dissolved in methanol at reflux (3200 mL). The solution was cooled to 25 °C and crystallized under stirring for two days. The suspension was filtered, washed with methanol and dried under vacuum for 18 h to yield a white solid product (170 g). A small part of this fraction (50 mg) was subjected to repeated chromatographic purifications by analytical HPLC-UV on a Polar-RP 80A 250 × 4.60 mm column. The mobile phase was a mixture of acetonitrile and water with 50 ppm (*v*/*v*) of formic acid added to both solvents with the following elution gradient: 0–5 min = H_2_O:CH_3_CN 85:15 isocratic; 6–22 min = from H_2_O:CH_3_CN 85:15 to H_2_O:CH_3_CN 70:30; 22–27 min = H_2_O:CH_3_CN 70:30 isocratic. Other parameters included injected volume 20 μL; flow rate 1.0 mL/min; UV detection λ 200 nm. Isomadecassoside (**4**, RT 24.5 min, 2.1 mg), asiaticoside B/terminoloside (**3**, RT 25.1 min, 3.8 mg) and madecassoside (**2**, RT 25.5 min, 25.9 mg) were obtained in pure states.

### 2.3. Biological Assays

#### 2.3.1. Cell Culture

J774A.1 macrophages (ATCC, from LGC Standards, Milan, Italy) were used for in vitro experiments. Cells were routinely maintained at 37 °C in a humidified incubator with 5% CO_2_ and were cultured in Dulbecco’s modified Eagle’s medium (DMEM, Lonza Group) supplemented with 10% fetal bovine serum (FBS), 100 U/mL penicillin and 100 μg/L streptomycin, 2 mM l-glutamine, 20 mM Hepes (4-(2-hydroxyethyl)-1-piperazineethanesulphonic acid) and 1 mM sodium pyruvate. The medium was changed every 48 h in conformity with the manufacturer’s protocols, and cell viability was evaluated by trypan blue exclusion.

#### 2.3.2. Nitrite Measurement and Pharmacological Treatment In Vitro

The anti-inflammatory effect of madecassoside (**2**), terminoloside (**3**) or isomadecassoside (**4**) was evaluated by measuring nitrite, stable metabolites of nitrite oxide (NO), in macrophages medium via a colorimetric assay, as previously described [16]. J774A.1 macrophages were seeded in 24-well plates (2.5 × 10^5^ cells per well) and incubated with the triterpene glycosides (2–50 µM) 30 min before LPS stimulation (1 μg/mL) for 24 h. Then, the cell supernatant was collected and incubated with 100 µL of Griess reagent (0.2% naphthylethylenediamine dihydrochloride and 2% sulphanilamide in 5% phosphoric acid) at room temperature for 10 min to allow the formation of a colored azo dye. The absorbance was read at 550 nm on a Thermo Scientific Multiskan GO instrument. Serial-diluted sodium nitrite (Sigma-Aldrich) was used to generate a standard curve. The data were expressed as µM of nitrite (n = 4 independent experiments including 3 replicates for each treatment).

#### 2.3.3. Cell Viability

The effect of madecassoside, terminoloside or isomadecassoside on cell viability was evaluated by measuring the incorporation of neutral red (NR), a weak cationic dye, in lysosomes (NR assay) [17].

## 3. Results and Discussion

### 3.1. Structural Elucidation of Isomadecassoside

A fraction obtained from *C. asiatica* leaves enriched in triterpenoid glycosides of the madecassoside series was subjected to reverse phase HPLC purification affording the new isomadecassoside (**4**), together with the known madecassoside (**2**) and terminoloside (asiaticoside B, **3**). The known compounds were readily identified by comparison of their physical and spectral data with those reported in the literature [10,11].

Compound 4 was isolated as a white amorphous solid with [*α*]^22^_D_—7.05 (*c* 12, CH_3_OH) and molecular formula C_48_H_78_O_20_ (HR-ESIMS found *m*/*z* 997.4978 [M + Na]^+^; C_48_H_78_O_20_Na requires 997.4984), the same as madecassoside.

The ^1^H NMR spectrum of 4 (Table 1) showed typical resonances of a triterpene glycoside including signals of one methyl doublet (*δ*_H_ 1.04) and five methyl singlets, four of which resonated upfield (*δ*_H_ 1.32, 1.29, 1.05, 1.00) and one at relatively low field (*δ*_H_ 1.65). An additional methyl doublet at *δ*_H_ 1.28 could be likely ascribable to a rhamnose sugar unit. In addition, the ^1^H NMR spectrum included a series of multiplets located between *δ*_H_ 0.87 and 2.50, all belonging to the aglycone moiety, signals of oxymethines and oxymethylenes located between *δ*_H_ 3.20 and 4.86, and two methines resonating as doublets at *δ*_H_ 5.26 and 5.39.

All the proton signals were associated to those of the directly linked carbon atoms through the correlations of the 2D NMR HSQC spectrum, which revealed the presence of only one olefinic proton (*δ*_H_ 5.26, *δ*_C_ 118.0) in the structure of **4**. A careful inspection of 2D NMR COSY, HSQC and HMBC spectra (Figure 3, see Appendix A) allowed the identification of the aglycone moiety of **4** as an ursane-type triterpene including three oxymethine protons (*δ*_H_ 4.37, 3.76, 3.30), one isolated oxymethylene (*δ*_H_ 3.58, 3.45) and an ester carbonyl at C-28 (*δ*_C_ 176.4). In particular, the five spin systems identified from the COSY spectrum, and highlighted in red in Figure 3, were connected through a network of key HMBC correlations (Figure 3). In particular, correlations from H_3_-24, H_3_-25, H_3_-26 and H_3_-27 (in black in Figure 3) were instrumental to build up the architecture of rings A-D, that proved to parallel the structure of madecassoside, with the single exception of a saturated C-12/C-13 bond.

As for ring E, the HMBC cross-peaks (in blue in Figure 3) of H_3_-30 (*δ*_H_ 1.65, s) with C-19 and the *sp^2^* C-20 (*δ*_C_ 144) and C-21, and those of H_3_-29 (*δ*_H_ 1.04, d, 8.0) with C-18, C-19 and C-20 (*δ*_C_144), were diagnostic for the presence of a trisubstituted double bond at Δ^20,21^. Moreover, the key HMBC correlations of H_2_-22 with C-18 (*δ*_C_ 49.4) and C-28 (*δ*_C_ 176.4) placed the ester carbonyl at C-28 and confirmed the planar structure of the aglycone of **4** as an ursene triterpenoid, sharing the same carbon framework of madecassoside but showing a different double bond location.

The almost complete superimposition of ^1^H/^13^C NMR signals and *J*_H-H_ coupling constants of **4** with those of madecassoside, supported by analysis of 2D NMR NOESY cross-peaks, strongly indicated that the two compounds shared the relative configuration of the common stereogenic centers, including the three oxymethines at C-2, C-3 and C-6. Moreover, the NOESY cross-peaks of H_3_-29 with H_3_-25 and H_3_-26 were indicative of the β-orientation of CH_3_-29, while the correlation H-13/H_3_-26 indicated the *trans* junction of C/D rings.

The ^1^H and ^13^C NMR resonances of three anomeric methines at *δ*_H_ 5.35, *δ*_C_ 95.0; *δ*_H_ 4.42, *δ*_C_ 104.2 and *δ*_H_ 4.86, *δ*_C_ 102.5, coupled through the HSQC spectrum, revealed the presence of three sugar moieties. The comprehensive and comparative analysis of the *J*_H-H_ coupling constant values and the detailed ^1^H- and ^13^C-NMR assignments based on the 2D NMR COSY, HSQC and HMBC spectra indicated that **4** shared with madecassoside the same sugar portion, including two β-glucopyranoses and one α-rhamnopyranose units linked as α-l-Rha-(1-4)-*O*-β-d-Glc-(1-6)-*O*-β-d-Glc. The HMBC cross-peak H-1’/C-28 supported the connection of the sugar and aglycone moieties through an ester linkage, thus establishing the structure of the new saponin isomadecassoside (**4**) as the O-α-l-rhamnopyranosyl-(1-4)-O-β-d-glucopyranosyl-(1-6)-O-β-d-glucopyranosyl ester of 2α,3β,6β,23-tetrahydroxyurs-20-en-28-oic acid.

The single *C. asiatica* saponin related to isomadecassoside is isoasiaticoside, reported in 2007 by Yu at al. [18] and belonging to the same urs-20-ene subtype, although its structure is dehydroxylated at position C-6. Thus, to the best of our knowledge, compound **4** is a triglycoside of the unprecedented pentacyclic triterpenoid 2α,3β,6β,23-tetrahydroxyurs-20-en-28-oic acid, for which we propose the trivial name isomadecassic acid.

### 3.2. Biological Activity

Won et al. reported an anti-inflammatory effect for madecassoside on LPS-stimulated RAW 264.7 murine macrophage cells with inhibition of NO production [15]. To compare the anti-inflammatory potency of the new saponin isomadecassoside (**4**) with the known madecassoside (**2**) and terminoloside (**3**), these three compounds were tested for their ability to reduce nitrite levels in LPS-stimulated macrophages J774A.1. LPS (1 μg/mL) treatment for 24 h caused a significant increase in nitrite levels (Figure 4), while a pre-treatment (30 min before LPS) with madecassoside (Figure 4A), terminoloside (Figure 4B) or isomadecassoside (Figure 4C) reduced LPS-induced nitrite production. At 50 µM, isomadecassoside (4) proved to be slightly but significantly more potent than its two analogues (19% reduction for isomadecassoside in place of 11% for madecassoside and terminoloside). At the higher concentration tested (i.e., 50 µM), compounds **2–4** did not affect cell vitality after 24 h exposure, thus excluding the possibility that the effect of the three compounds on nitrite production could be due to a non-specific cytotoxic effect in macrophages (data not shown).

## 4. Conclusions

In conclusion, our study revealed that, in addition to the oleanane analogue terminoloside, the madecassoside fraction of the *C. asiatica* extract also contains significant amounts of the new saponin isomadecassoside (**4**). This compound is a triglycoside ester of an unprecedented ursane acid, namely 2α,3β,6β,23-tetrahydroxyurs-20-en-28-oic acid (isomadecassic acid). Since isomadecassoside (**4**) shares the same molecular formula and a closely similar chromatographic behavior with madecassoside (**2**) and terminoloside (**3**), it had been overlooked in the dozens of phytochemical analyses on *C. asiatica* carried out to date. Thus, our work indicates that the madecassoside chromatographic peak is a trio and not a duet (madecassoside/terminoloside), as believed.

From the pharmacological point of view, the nitrite reduction potential seems almost uniform within this trio of compounds, with isomadecassoside showing a similar (if not better) activity than madecassoside.

## Figures and Tables

**Figure 1 biomolecules-11-00494-f001:**
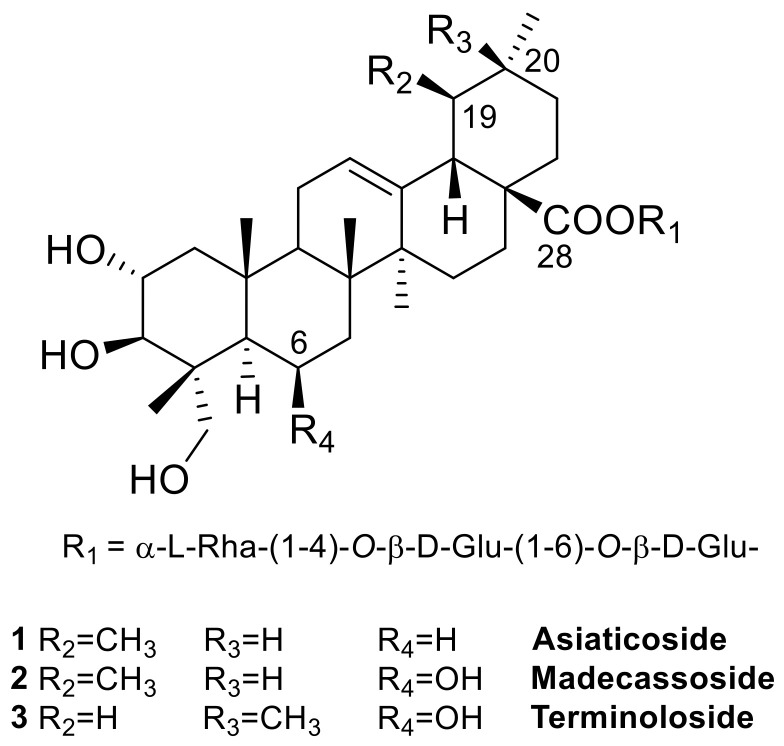
The main triterpene saponins isolated from *C. asiatica*.

**Figure 2 biomolecules-11-00494-f002:**
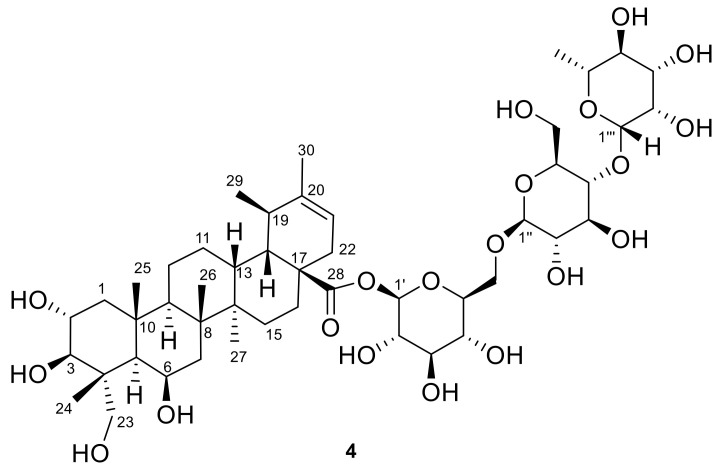
The chemical structure of the new isomadecassoside (**4**).

**Figure 3 biomolecules-11-00494-f003:**
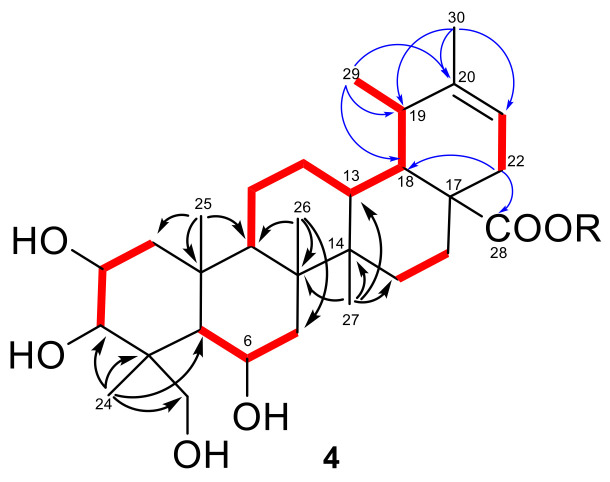
COSY (in red bold) and key H→C HMBC (black/blue arrows) correlations detected for **4**.

**Figure 4 biomolecules-11-00494-f004:**
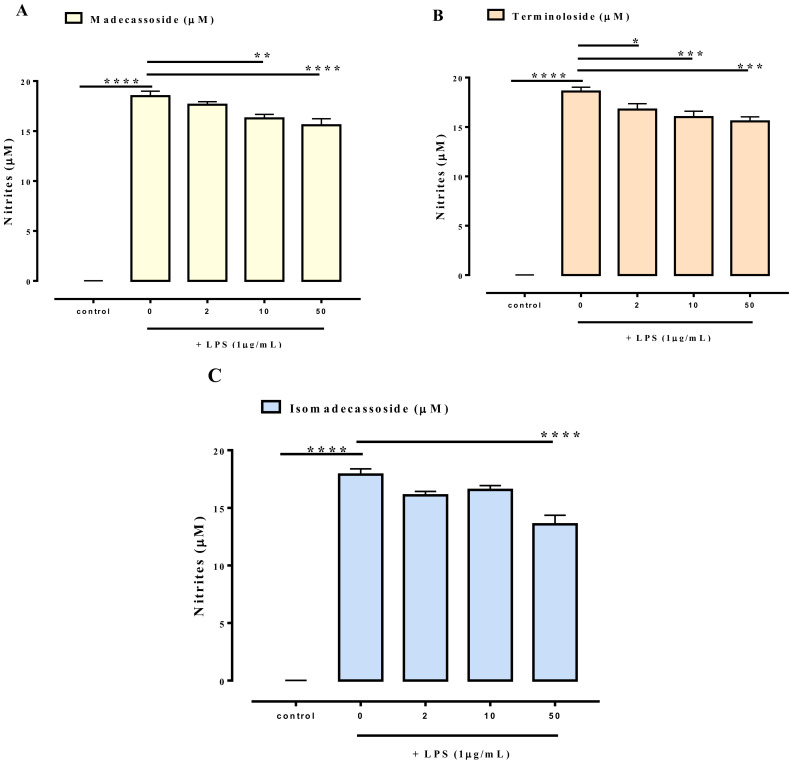
Inhibitory effect of madecassoside (**A**), terminoloside (**B**) and isomadecassoside (**C**) on nitrite levels in the cell medium of J774A.1 macrophages stimulated with lipopolysaccharide (LPS, 1 μg/mL) for 24 h. The compounds were added to the cell media 30 min before LPS stimulus. Results are expressed as mean ± SEM of four independent experiments (in quadruplicate). * *p* < 0.05, ** *p* < 0.01, *** *p* < 0.001 and **** *p* < 0.0001.

**Table 1 biomolecules-11-00494-t001:** ^1^H (700 MHz) and ^13^C (175 MHz) NMR data of compound **4** in CD_3_OD.

Pos.	*δ*_H_, Mult., *J* in Hz	*δ*_C_, Type
1a	2.00, dd, 12.5, 5.0	49.9, CH_2_
1b	0.87, dd, 12.5, 11.5	
2	3.76, ddd, 11.5, 8.1, 5.0	69.6, CH
3	3.30 ^a^	77.7, CH
4		44.7, C
5	1.23 ^a^	48.9, CH
6	4.37, m	68.5, CH
7a	1.65 ^a^	42.0, CH_2_
7b	1.61 ^a^	
8		42.0, C
9	1.50 ^a^	52.0, CH
10		38.7, C
11a	1.51 ^a^	22.7, CH_2_
11b	1.47 ^a^	
12a	1.80 ^a^	28.5, CH_2_
12b	1.21 ^a^	
13	2.50, ddd, 13.3, 12.3, 2.9	39.7, CH
14		42, C
15a	1.59 ^a^	30.2, CH_2_
15b	1.15 ^a^	
16a	2.06, dt, 12.6, 3.0	33.7, CH_2_
16b	1.45 ^a^	
17		50.4, C
18	1.28 ^a^	49.4, CH
19	2.17, m	38.3, CH
20		144.0, C
21	5.26, d, 7.1	118.0, CH
22a	2.28, dd, 15.2, 7.1	38.3, CH_2_
22b	1.85 ^a^	
23a	3.58, d, 11.2	65.5, CH_2_
23b	3.45, d, 11.2	
24	1.05, s	14.7, CH_3_
25	1.32, s	19.7, CH_3_
26	1.29, s	17.2, CH_3_
27	1.00, s	15.2, CH_3_
28		176.4, C
29	1.04, d, 8.0	23.6, CH_3_
30	1.65, bs	21.9, CH_3_
O-*β*-d-Glc (first)		*(continued)*
1′	5.39, d, 8.2	95.0, CH
2′	3.31 ^a^	73.9, CH
3′	3.41 ^a^	77.9, CH
4′	3.63, dd, 9.5, 3.4	71.8, CH
5′	3.53 ^a^	79.4, CH
6a′	4.09, dd, 12.0, 1.5	69.2, CH_2_
6b′	3.80, dd, 12.0, 5.4	
O-*β*-d-Glc (second)		
1″	4.42, d, 8.0	104.2, CH
2″	3.24, t, 8.3	75.3, CH
3″	3.47 ^a^	76.4, CH
4″	3.54 ^a^	79.4, CH
5″	3.35 ^a^	76.5, CH
6a″	3.84 ^a^	61.8, CH_2_
6b″	3.67, dd, 12.3, 4.7	
O-*α*-l-Rha		
1‴	4.86 ^a^	102.5, CH
2‴	3.84 ^a^	72.1, CH
3‴	3.31 ^a^	73.2, CH
4‴	3.42 ^a^	73.4, CH
5‴	3.96, m	70.4, CH
6‴	1.28, d, 6.1	17.5, CH_3_

^a^ Overlapped with other signals.

## Data Availability

Not applicable.

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
