# Peer review of "Isomadecassoside, a New Ursane-Type Triterpene Glycoside from Centella asiatica Leaves, Reduces Nitrite Levels in LPS-Stimulated Macrophages"

_biomolecules, 2021, doi:10.3390/biom11040494_

Round 1

Reviewer 1 Report

The manuscript describes isolation and structure elucidation of a novel triterpene glycoside, isomadecassoside, from Centella asiatica. The authors may consider only a few points to improve the manuscript. The authors need to show how to collect (or purchase) the plant leaves because triterpene composition of different sources of this plant varies considerably (according to Reference 3). The authors also need to show detailed information about obtaining a madecassoside and terminoloside rich fraction (e.g. how to prepare an extract and the crude fraction).

Author Response

The authors need to show how to collect (or purchase) the plant leaves because triterpene composition of different sources of this plant varies considerably (according to Reference 3). The authors also need to show detailed information about obtaining a madecassoside and terminoloside rich fraction (e.g. how to prepare an extract and the crude fraction).

Answer: The reviewer is right. This information was needed. We have added in par. 2.2. the origin of leaves (Madagascar) and detailed procedure from leaves to madecassoside and terminoloside rich fraction.  

Reviewer 2 Report

Giuseppina et. al studied on “Isomadecassoside, a New Anti-inflammatory Triterpene Glycoside from Centella asiatica”.  This manuscript needs major revision for publication.

  • What is anti-inflammatory activity? The study is not available in manuscript.
  • The author needs to provide mechanism for anti-inflammatory effect for madecassoside.
  • The author should provide strong data to support Inhibitory effect of madecassoside.
  • How chromatographic behaviour with madecassoside conducted?
  • There is not much description about differences in madecassoside, terminoloside and isomadecassoside
  • Author must provide chemicals used in the experiment. I did not find this term in the whole manuscript.
  • Ultimately, it fails to design advantages and the potential application of Anti-inflammatory Triterpene Glycoside from Centella asiatica.

Author Response

What is anti-inflammatory activity? The study is not available in manuscript. The author needs to provide mechanism for anti-inflammatory effect for madecassoside. The author should provide strong data to support Inhibitory effect of madecassoside.

Answer: We agree with reviewer that this is not an anti-inflammatory study but an evaluation of  the inhibitory effect of madecassoside (A), terminoloside (B) and isomadecassoside (C) on nitrite levels in macrophages stimulated with lipopolysaccharide. The two things can be related but they do not coincide.

Thus, the title has been changes as follows: Isomadecassoside, a New Ursane-Type Triterpene Glycoside from Centella asiatica Leaves, Reduces Nitrite Levels in LPS-Stimulated Macrophages.

The abstract has also been changed deleting the mention to anti-inflammatory activity. The term anti-inflammatory has been deleted also from Conclusions.

How chromatographic behaviour with madecassoside conducted? There is not much description about differences in madecassoside, terminoloside and isomadecassoside

Answer: We have added several more information about the chromatographic purification in par. 2.2

Author must provide chemicals used in the experiment. I did not find this term in the whole manuscript.

Answer: Origin of chemicals and solvents has been added at the end of par. 2.1

Ultimately, it fails to design advantages and the potential application of Anti-inflammatory Triterpene Glycoside from Centella asiatica.

Answer: The aim of our manuscript, now more clearly expressed in the title, was to report on the isolation of a new triterpene glycoside from the medicinal plant Centella asiatica and on the comparison of its nitrite-reduction activity with that of madecassoside.

The plant extract already finds wide application and there are several formulations in the market, including those from Indena SpA, co-author of this study. Thus, we believe that disclosing and characterizing a further component of C. asiatica extract is an important information for the scientific community.    

Reviewer 3 Report

The manuscript biomolecules-1149418, “Isomadecassoside, a New Anti-inflammatory Triterpene Glycoside from Centella asiatica”, has a good scientific value, originality and is well organized. I have just some small suggestions that would improve its quality.

I suggest the authors to change the title. The authors tested the inhibitory effect of madecassoside (A), terminoloside (B) and isomadecassoside (C) on nitrite levels in macrophages stimulated with lipopolysaccharide (see the legend of figure 4). Even if this assay is correlated with the inflammatory reactions, they are not the same thing. Usually the anti-inflammatory effect is tested directly in animals and related to clinical aspects of inflammation. I recommend the authors to choose a title that describes better their work: the discovery of a new derivate of the ursane acid previously overlooked in phytochemical analyses on C. asiatica and its effect to slightly reduce the nitrite levels in macrophages stimulated with lipopolysaccharide.

row 30, indicate if the whole plant or just the leaves are used for those indications

row 45, even if is generally known, explain what Glu, and Rha mean

Row 48, the authors mention “C-6 dehydroxylated analogue madecassoside (2)”, but in figure 1 the compound 2 has a hydroxyl as R4.

row 49, the compound 3, terminoloside, is mentioned. Please check!

Row 57, add the dose of asiaticoside, the duration of treatment, and the type of animal (mice or rats?)

Row 60, add the dose, and detail the cancer cells used.

row 63, even if is generally known, explain what LPS means

row 66, remove the self-citations. The mentioned research is not related and add not relevant information for this paper

rows 109-112 this section represents results, and not methods. It should be moved accordingly

row 127 and row 120, 24h or 24 h?

row 146, compound 4 is presented as C48H78O20, but on row 111 as C48H78O20Na

row 231, the quality of the figure should be improved. It is very hard to understand it.

The section on results should be divided in analytical and biological subsections

In the section 213-220 the authors should present the percentage of reduction of nitrite levels. The readers should observe that the effect is not very impressive as the authors seem to imply. If we would extrapolate and estimate an IC50, that IC50 value would be quite high. As least over 100 μM. Not very promising in terms of clinical usefulness.

Author Response

The manuscript biomolecules-1149418, “Isomadecassoside, a New Anti-inflammatory Triterpene Glycoside from Centella asiatica”, has a good scientific value, originality and is well organized. I have just some small suggestions that would improve its quality. I suggest the authors to change the title. The authors tested the inhibitory effect of madecassoside (A), terminoloside (B) and isomadecassoside (C) on nitrite levels in macrophages stimulated with lipopolysaccharide (see the legend of figure 4). Even if this assay is correlated with the inflammatory reactions, they are not the same thing. Usually the anti-inflammatory effect is tested directly in animals and related to clinical aspects of inflammation. I recommend the authors to choose a title that describes better their work: the discovery of a new derivate of the ursane acid previously overlooked in phytochemical analyses on C. asiatica and its effect to slightly reduce the nitrite levels in macrophages stimulated with lipopolysaccharide.

Answer: We agree with reviewer. The title has been changes as follows: Isomadecassoside, a New Ursane-Type Triterpene Glycoside from Centella asiatica Leaves, Reduces Nitrite Levels in LPS-Stimulated Macrophages.

The abstract has also been changed deleting the mention to anti-inflammatory activity. The term anti-inflammatory has been deleted also from Conclusions.

Row 30, indicate if the whole plant or just the leaves are used for those indications

Answer: Done

row 45, even if is generally known, explain what Glu, and Rha mean

Answer: Done

Row 48, the authors mention “C-6 dehydroxylated analogue madecassoside (2)”, but in figure 1 the compound 2 has a hydroxyl as R4.

Answer: Corrected. it was "hydroxylated"

row 49, the compound 3, terminoloside, is mentioned. Please check!

Answer: Terminoloside is mentioned in the following paragraph

Row 57, add the dose of asiaticoside, the duration of treatment, and the type of animal (mice or rats?)

Answer: This information has been provided

Row 60, add the dose, and detail the cancer cells used.

Answer: This information has been provided

row 63, even if is generally known, explain what LPS means

Answer: Done

row 66, remove the self-citations. The mentioned research is not related and add not relevant information for this paper

Answer: The three references have been removed

rows 109-112 this section represents results, and not methods. It should be moved accordingly

Answer: This information has been moved in Results and Discussion

row 127 and row 120, 24h or 24 h?

Answer: Corrected

row 146, compound 4 is presented as C48H78O20, but on row 111 as C48H78O20Na

Answer: In HR-ESIMS the compound is revealed as its sodium adduct

row 231, the quality of the figure should be improved. It is very hard to understand it.

Answer: The graphs have been repositioned and the size increased to make the figure more readable.

The section on results should be divided in analytical and biological subsections

Answer: We have identified two subsections in the Results and Discussion section

In the section 213-220 the authors should present the percentage of reduction of nitrite levels. The readers should observe that the effect is not very impressive as the authors seem to imply. If we would extrapolate and estimate an IC50, that IC50 value would be quite high. As least over 100 μM. Not very promising in terms of clinical usefulness.

Answer: The percentage of reduction have been added.  

Round 2

Reviewer 2 Report

Your manuscript is accepted.